# Effects of Grazing, Extreme Drought, Extreme Rainfall and Nitrogen Addition on Vegetation Characteristics and Productivity of Semiarid Grassland

**DOI:** 10.3390/ijerph20020960

**Published:** 2023-01-05

**Authors:** Jing Zhang, Xiaoan Zuo, Peng Lv

**Affiliations:** Northwest Institute of Eco-Environmental and Resources, Chinese Academy of Sciences, Lanzhou 730000, China

**Keywords:** grazing, rainfall changes, nitrogen addition, vegetation characteristics, soil water and nitrogen content, productivity, structural equation model

## Abstract

Grassland use patterns, water and nutrients are the main determinants of ecosystem structure and function in semiarid grasslands. However, few studies have reported how the interactive effects of rainfall changes and nitrogen deposition influence the recovery of semiarid grasslands degraded by grazing. In this study, a simulated grazing, increasing and decreasing rainfall, nitrogen deposition test platform was constructed, and the regulation mechanism of vegetation characteristics and productivity were studied. We found that grazing decreased plant community height (CWM_height_) and litter and increased plant density. Increasing rainfall by 60% from May to August (+60%) increased CWM_height_; decreasing rainfall by 60% from May to August (–60%) and by 100% from May to June (−60 d) decreased CWM_height_ and coverage; −60 d, +60% and increasing rainfall by 100% from May to June (+60 d) increased plant density; −60% increased the Simpson dominance index (D index) but decreased the Shannon–Wiener diversity index (H index); −60 d decreased the aboveground biomass (ABG), and −60% increased the underground biomass (BGB) in the 10–60 cm layer. Nitrogen addition decreased species richness and the D index and increased the H index and AGB. Rainfall and soil nitrogen directly affect AGB; grazing and rainfall can also indirectly affect AGB by inducing changes in CWM_height_; grazing indirectly affects BGB by affecting plant density and soil nitrogen. The results of this study showed that in the semiarid grassland of Inner Mongolia, grazing in the nongrowing season and grazing prohibition in the growing season can promote grassland recovery, continuous drought in the early growing season will have dramatic impacts on productivity, nitrogen addition has a certain impact on the species composition of vegetation, and the impact on productivity will not appear in the short term.

## 1. Introduction

Grazing interactions with human activities and climate change have greatly affected rainfall patterns and the biogeochemical cycles of grassland [1]. These global change drivers have significantly impacted plant productivity, biodiversity and ecosystems [2,3]. In particular, arid and semiarid grasslands may be particularly susceptible to rainfall variations and soil nutrients, besides having limited grazing [4,5].

Grazing is one of the most common uses of grasslands. Overgrazing results in simplified community composition and reduced productivity [6], leading to the destruction of vegetation canopies and surface lichens and increasing the risk of soil erosion [7]. Plant coverage, density and aboveground biomass are significantly enhanced in the early stage of fencing, but this enhancement is weakened or even reversed with the extension of fencing time [8]. The moderate-disturbance hypothesis posits that moderate-intensity grazing can improve species diversity and the productivity of grassland, which is beneficial to the health and sustainable development of grassland ecosystems [9,10]. Research has also shown that the quantitative characteristics of grassland communities are influenced by grazing during the growing months, but the effects are offset by nongrowing-season grazing [11]. Therefore, grazing patterns may greatly affect grassland biomass and soil environment, and grazing during the nongrowing season is beneficial for improving the quality of grasslands [12].

The response mechanism of grassland vegetation to extreme climate has become a major problem in grassland-adaptation research. The 2011 IPCC report clearly points out that global climate change is likely to increase the intensity and frequency of extreme climate events, such as severe drought and heavy precipitation, at global and regional levels in the future [13], which is mainly reflected in significant changes in the total amount and distribution of rainfall [14,15]. Drought can inhibit plant growth and reduce coverage, diversity and productivity [16,17], affecting the persistence and distribution of plant species and negatively affecting seedling survival [18]. Increasing rainfall can promote soil microbial activities, change soil-nitrogen-use efficiency and ultimately improve productivity [19]. In addition to total rainfall, rainfall patterns can affect underground biomass increase in the growing season, which gradually decreases at the end of the growing season [20]. Therefore, the amount and pattern of multiple rainfall pulses during the growing season are crucial for regulating the structure, composition and productivity of water-constrained grassland ecosystems.

In the last three decades, the amount of atmospheric nitrogen deposition in China has closely followed that in North America and Western Europe, and China now has the third-largest concentration of nitrogen deposition in the world [21,22]. Studies on grasslands around the world have shown that nitrogen addition can promote plant growth by increasing soil-nitrogen availability and plant water-use efficiency [23], which are key factors affecting productivity [24,25]. In addition, nitrogen addition can reduce species richness and improve community biomass by increasing leaf size and plant height [26,27]. However, nitrogen deposition can also lead to soil acidification, which indirectly affects plant growth and changes the structure and function of grassland ecosystems [28,29]. In water-restricted areas, water and nitrogen play vital roles in plant growth and development [30]. The coupling effects of water and nitrogen also represent a mutual compensation mechanism for plant growth [31]. Ongoing changes in rainfall variability and nitrogen availability accelerate nutrient cycling in initial soil-nitrogen-mineralization rates and microbial activity, and finally change the plant-community composition and net primary productivity [32,33]. Whether water and nitrogen produce an enrichment effect depends more on interannual variations in precipitation conditions caused by naturally occurring extreme rainfall events [34].

Vegetation is an indispensable part of grassland ecosystems and the material basis of material circulation and energy flow [35]. Growth characteristics and species diversity can reflect the structure and function of plant communities [36], and species in different communities have different responses to external environmental changes. In semiarid grasslands in China, grazing is the most important mode of grassland utilization, and water and nitrogen are the most influential factors dictating plant growth [37,38]. Therefore, in the context of climate changes and increasing human activities, revealing the influencing mechanisms of grazing, rainfall changes, nitrogen addition and their interactions with vegetation characteristics and soil water and nitrogen content will not only help to predict variation trends of semiarid grassland ecosystems in the future but will also be of scientific significance in curbing grassland degradation and promoting the sustainable development of grassland ecosystems.

Horqin Sandy Land is a typical farming–pastoral ecotone in a semiarid area of northern China. Due to climate change and the intensification of agricultural activities in the last half-century, the vegetation structure in this area has changed greatly and productivity has decreased, which has resulted in this area exhibiting the most serious desertification [39]. However, there are still some questions that remain unclear. For instance, how do grazing, rainfall changes, nitrogen addition and their interactions affect vegetation and soil characteristics in sandy grassland? What are the regulatory mechanisms of sandy grassland productivity under grazing, rainfall changes, nitrogen addition and their interactions? In order to solve the above problems, this study selected the sandy grassland in Horqin Sandy Land as the object, established the field in situ simulation control experiment, and studied the dynamic changes of vegetation characteristics, soil water and soil nitrogen content in Horqin sandy grassland under grazing, rainfall changes, nitrogen addition and their interaction. This study is of great significance for the restoration and management of semiarid degraded grassland ecosystems.

## 2. Materials and Methods

### 2.1. Study Area

The study area is located in the territory of Naiman Banner, Tongliao City, Inner Mongolia in the south-central part of the Horqin Sandy Land (42°55′–42°57′ N, 120°40′–120°43′ E) (Figure 1). The area is about 360 m above sea level, and the average annual temperature is about 6.4 °C. The annual rainfall is about 360 mm, of which 70–80% is concentrated in June to August [40]. The dominant annual species in the sandy grassland are *Tribulus terretris*, *Setaria viridis* and *Artemisia scoparia*, and the dominant perennial species in the sandy grassland are *Pennisetum centrasiaticum*, *Cleistogenes squarrosa* and *Lespedeza bicolor*. The soil in the study area is aeolian sandy soil, according to the Chinese soil taxonomy classification system (http://www.resdc.cn) (accessed on 10 September 2021).

### 2.2. Experimental Design and Measuring

This study was conducted in 2018 and selected the sandy grassland that had begun fencing grazing at the Naiman Desertification Research Station in 2015 as the research object (120°42′ E, 42°55′ N). In addition to the grazing factor, the experimental design includes two factors, each of which is treated as follows:

Based on long-term observational data of total annual rainfall, extreme rainfall and extreme drought events in the growing season in the study region [14], we set up four rainfall treatments. ±60%: increasing or decreasing rainfall by 60% throughout the growing season (May to August); ±60 d: increasing or decreasing rainfall by 100% in the early growing season (May to June). Rainfall-increasing and -decreasing devices are widely used in climate change research because of their low cost and minimal climate impact [41] (Figure 2).

In 2018, the total rainfall in the early growing season (May to June) was 106.08 mm, and the total rainfall in the late growing season (July to August) was 270.28 mm. The total rainfall was 150.55 mm under the −60% treatment; the total rainfall was 602.18 mm under the +60% treatment; the total rainfall was 270.30 mm under the −60 d treatment; the total rainfall was 482.44 mm under the +60 d treatment (Table 1).

The Horqin Sandy Land is located in the ecologically fragile area of the semiarid agricultural and pastoral staggered areas in northern China, where large-scale human activities such as farmland fertilization and animal husbandry have intensified nitrogen imports, and as a result, the nitrogen load in the area is too high [42]. With reference to the nitrogen sedimentation levels of other countries in the world (such as the United States and Europe), adding 20g of nitrogen/m^2^ is a higher nitrogen sedimentation level in China [27]. Therefore, this experiment adds 10 g nitrogen/m^2^ in May and July each year, for a total of 20 g nitrogen/m^2^.

In this experiment, a three-factor full-factor randomized block design was adopted, and 48 large samples of 8 m × 8 m were first selected, and then 24 samples were randomly selected for rainfall-increasing and -decreasing experiments, and the sample area of each rainfall treatment was 2.5 m × 2.5 m. The experiment consisted of a total of 20 treatments, each with 6 replicates for a total of 120 quadrats (Figure 3).

### 2.3. Vegetation Characteristics Investigation and Soil Water and Nitrogen Content Determination

In this study, 120 small quadrats with an area of 1 m × 1 m were randomly set up in the sample plot to investigate vegetation characteristics and soil water and nitrogen content. Vegetation characteristic indicators including plant community height (CWM_height_), coverage, species richness, plant density, litter biomass, Simpson dominance index (D index), the Shannon–Wiener diversity index (H index), Pielou evenness index (J index), aboveground biomass (AGB) and belowground biomass (BGB) were determined. In detail, the total coverage of the vegetation within the sample and the sub-coverage of each species were estimated via visual inspection. The maximum height of each plant species (the natural vertical distance from the top of the plant to the ground) was determined with a tape measure, and the number of plants of each species was recorded during this process, and then we harvested the litter in the samples. All collected samples were loaded into corresponding paper envelopes and dried to a constant weight in an oven at 65 °C to obtain AGB and litter biomass. Finally, two soil samples were drilled in layers in each sample square with a diameter of 10 cm, and the soil layer distribution was 0–10 cm, 10–20 cm, 20–40 cm and 40–60 cm. They were then mixed and installed in a nylon mesh bag after air-drying through a sieve with a pore size of 2 mm to remove impurities. The soil samples were dried to a constant weight (105 °C, 24 h), and soil water content was obtained. Determination of soil nitrogen content was performed using an elemental analyzer (Costech Ecs4010, Florence, Italy).

### 2.4. Methods for Measuring Plant Species Diversity

In the below formula, the relative height is the maximum height of a particular plant species divided by the sum of the maximum heights of all plants in the quadrats with an area of 1 m × 1 m. The relative coverage is the coverage of a plant species divided by the sum of the coverage of all plant species in the quadrats with an area of 1 m × 1 m. The relative biomass is the biomass of a particular plant species divided by the sum of the biomass of all plant species in the quadrats with an area of 1 m × 1 m.
Plant species importance = (Relative height + Relative coverage + Relative biomass)/3(1)

The diversity index of grassland vegetation was calculated according to the species composition of plant communities, and the diversity measurement index was reflected by the Simpson dominance index (D index), the Shannon–Wiener diversity index (H index) and the Pielou evenness index (J index). The species richness is expressed as the number of species that appear within a 1 m × 1 m sample. The formula for the calculation is as follows:(2)$$D=\sum_{i}^{S}{N_{i}}^{2}$$
(3)$$H=\sum_{i}^{S}N_{i}\mathrm{Ln}N_{i}$$
*J* = *H*/ln*S*(4)

In the above formula: *N_i_* is the relative importance value of species *i*, and *S* is the number of species in the community.

### 2.5. Plant Community Height (*CWM_height_*)

Plant community height (CWM_height_) is the sum of the maximum height of each plant in the community multiplied by its relative biomass, reflecting the overall height of the community, calculated by the following equation:(5)$$\mathrm{CWM}=\sum_{i=1}^{n}P_{i}\times {trait}_{i}$$

In the above formula: *P* is the relative biomass of species *i* within the community, and trait is the height of species *i*.

### 2.6. Data Analysis

The statistics in the paper were all mean ± standard errors. The effects of grazing, rainfall changes and nitrogen addition and their interactions on vegetation characteristics and soil water and nitrogen content in the sandy grassland were analyzed using three-factor variance, and the effects of grazing, rainfall changes, nitrogen addition and soil depth on BGB in sandy grassland were analyzed using four-factor variance. The correlation between vegetation characteristics and soil water and nitrogen content was analyzed by the Pearson correlation analysis method. The multiple comparisons were tested for differences in significance at the *p* < 0.05 and *p* < 0.01 levels using the Least Significant Difference Method (LSD). The data analysis was performed using SPSS 21.0 (IBM Systat Software, New York, NY, USA), SigmaPlot 12.5 (Systat Software, San Jose, CA, USA) was used for drawing, and the AMOS 20.0 (IBM Systat Software, New York, NY, USA) was used for structural equation modeling (Data are presented in Appendix A).

## 3. Results and Analysis

### 3.1. Plant Importance Values of Sandy Grassland under Fencing under Rainfall Changes, Nitrogen Addition and Their Interaction

In the fencing grassland, under CK × N0 treatment, the dominant species were *Cleistogenes squarrosa*, *Setaria viridis* and *Artemisia scoparia*, and the significance values, respectively, were 30.61%, 20.38% and 13.11%. Under CK × N0 treatment, the dominant species were *Tribulus terretris*, *Cleistogenes squarrosa* and *Chenopodium acuminatum*, and the significance values, respectively, were 21.80%, 20.86% and 14.22%. Under −60% × N0 treatment, the dominant species were *Cleistogenes squarrosa*, *Tribulus terretris* and *Chenopodium acuminatum*, and the significance values, respectively, were 23.43%, 19.48% and 17.91%. Under −60% × +N treatment, the dominant species were *Pennisetum centrasiaticum*, *Tribulus terretris* and *Cleistogenes squarrosa*, and the significance values, respectively, were 39.74%, 24.03% and 23.00%. Under +60% × N0 treatment, the dominant species were *Lespedeza bicolor*, *Pennisetum centrasiaticum* and *Setaria viridis*, and the significance values, respectively, were 25.88%, 24.81% and 21.03%. Under +60% × +N treatment, the dominant species were *Pennisetum centrasiaticum*, *Setaria viridis* and *Tribulus terretris*, and the significance values, respectively, were 37.65%, 23.57% and 16.39%. Under −60 d × N0 treatment, the dominant species were *Cleistogenes squarrosa*, *Tribulus terretris* and *Pennisetum centrasiaticum*, and the significance values, respectively, were 23.96%, 22.53% and 18.19%. Under −60 d × +N treatment, the dominant species were *Cleistogenes squarrosa*, *Setaria viridis* and *Tribulus terretris*, and the significance values, respectively, were 27.46%, 19.87% and 16.33%. Under +60 d × N0 treatment, the dominant species were *Setaria viridis*, *Cleistogenes squarrosa* and *Artemisia scoparia*, and the significance values, respectively, were 27.05%, 26.92% and 15.81%. Under +60 d × +N treatment, the dominant species were *Cleistogenes squarrosa*, *Setaria viridis* and *Pennisetum centrasiaticum*, and the significance values, respectively, were 23.04%, 21.50% and 21.16% (Figure 4).

### 3.2. Plant Importance Values of Sandy Grassland under Grazing under Rainfall Changes, Nitrogen Addition and Their Interaction

In the grazing grassland, under CK × N0 treatment, the dominant species were *Cleistogenes squarrosa*, *Artemisia scoparia* and *Setaria viridis*, and the significance values, respectively, were 37.05%, 14.73% and 12.15%. Under CK × +N treatment, the dominant species were *Tribulus terretris*, *Cleistogenes squarrosa* and *Chenopodium acuminatum*, and the significance values, respectively, were 33.81%, 22.64% and 14.65%. Under −60% × N0 treatment, the dominant species were *Tribulus terretris*, *Cleistogenes squarrosa* and *Setaria viridis*, and the significance values, respectively, were 31.47%, 19.73% and 13.99%. Under −60% × +N treatment, the dominant species were *Tribulus terretris*, *Cleistogenes squarrosa* and *Pennisetum centrasiaticum*, and the significance values, respectively, were 34.61%, 22.98% and 15.93%. Under +60% × N0 treatment, the dominant species were *Pennisetum centrasiaticum*, *Artemisia scoparia* and *Setaria viridis*, and the significance values, respectively, were 23.35%, 22.21% and 20.85%. Under +60% × +N treatment, the dominant species were *Tribulus terretris*, *Setaria viridis* and *Cleistogenes squarrosa*, and the significance values, respectively, were 28.52%, 22.17% and 18.82%. Under −60 d × N0 treatment, the dominant species were *Cleistogenes squarrosa*, *Setaria viridis* and *Tribulus terretris*, and the significance values, respectively, were 24.09%, 19.19% and 17.61%. Under −60 d × +N treatment, the dominant species were *Tribulus terretris*, *Setaria viridis* and *Pennisetum centrasiaticum*, and the significance values, respectively, were 28.06%, 22.35% and 20.23%. Under +60 d × N0 treatment, the dominant species were *Cleistogenes squarrosa*, *Pennisetum centrasiaticum* and *Artemisia scoparia*, and the significance values, respectively, were 26.01%, 17.66% and 14.25%. Under+60 d × +N treatment, the dominant species were *Setaria viridis*, *Chenopodium acuminatum* and *Tribulus terretris*, and the significance values, respectively, were 27.02%, 19.31% and 18.98% (Figure 5).

### 3.3. Effects of Grazing, Rainfall Changes, Nitrogen Addition and Their Interactions on Vegetation Characteristics in the Sandy Grassland

Grazing had extremely significant effects on CWM_height_, plant density and litter (*p* < 0.01) and had significant effects on Simpson dominance index (*p* < 0.05). Rainfall changes had extremely significant effects on CWM_height_, coverage, Simpson dominance index, the Shannon–Wiener diversity index and AGB (*p* < 0.01) and had significant effects on plant density (*p* < 0.05). Nitrogen addition had extremely significant effects on species richness, Simpson dominance index and the Shannon–Wiener diversity index (*p* < 0.01) and had significant effects on AGB (*p* < 0.05). The interaction between grazing and rainfall had a significant effect on CWM_height_ (*p* < 0.05) (Table 2).

Grazing decreased CWM_height_ and litter, but increased plant density (*p* < 0.05). In the fencing grassland, −60% and −60 d treatments decreased CWM_height_ (*p* < 0.05). In the grazing grassland, +60% and +60 d treatment increased CWM_height_ (*p* < 0.05). Under the −60% and CK treatments, grazing decreased CWM_height_ (*p* < 0.05) (Figure 6).

The −60% and −60 d treatments decreased CWM_height_ and coverage (*p* < 0.05). The −60 d, +60 d and +60% treatments increased plant density (*p* < 0.05). The −60% treatment increased Simpson dominance index but decreased the Shannon–Wiener diversity index (*p* < 0.05) (Figure 7).

The −60 d treatment decreased AGB (*p* < 0.05). Rainfall changes had no significant effect on BGB in the 0–10 cm soil layer, while the −60% treatment increased the BGB at soil depths of 10–20 cm, 20–40 cm and 40–60 cm (*p* < 0.05) (Figure 8).

Nitrogen addition decreased species richness and Simpson dominance index (*p* < 0.05) but increased the Shannon–Wiener diversity index and AGB (*p* < 0.05) (Figure 9).

### 3.4. Effects of Grazing, Rainfall Changes, Nitrogen Addition on Productivity in the Sandy Grassland

On the basis of correlation analysis, structural equation model was established to quantify the direct and indirect effects of grazing, rainfall and nitrogen addition on vegetation characteristics, soil water and nitrogen content and productivity of sandy grassland. Models based on grazing, rainfall, soil nitrogen and CWM_height_ provided the best explanation for AGB (χ^2^ = 0.51, *p* = 0.98. RMSEA = 0.00. GFI = 0.95), explaining 51% of AGB in total (Figure 10a). Models based on grazing, plant density and soil nitrogen provided the best explanation for BGB (χ^2^ = 0.23, *p* = 0.63; RMSEA = 0.00; GFI = 0.95), explaining 21% of BGB in total (Figure 10b). The structural equation model showed that rainfall could directly affect AGB and indirectly affect AGB through CWM_height_, grazing could indirectly affect AGB by affecting CWM_height_, and soil nitrogen could directly affect AGB. Plant density and soil nitrogen could directly affect BGB, while grazing could indirectly affect BGB by affecting plant density and soil nitrogen.

## 4. Discussion

The importance values reflect the dominant position and function of plant species in the community [43]. In our study, grazing increased the importance value of annuals such as *Chenopodium acuminatum*, *Tribulus terretris* and *Eragrostis pilosa*. Rainfall changes and nitrogen addition also affected the importance values of plants in the sandy grassland. Increased rainfall decreased the dominance of *Cleistogenes squarrosa*, while decreased rainfall increased the dominance of *Tribulus terretris* and *Salsolacollina*. Nitrogen addition enhanced the dominance of *Chenopodium acuminatum*, *Tribulusterretris* and *Pennisetum centrasiaticum*, but decreased the dominance of *Setaria viridis*, *Artemisia scoparia* and *Cleistogenes squarrosa*. This indicates that different species have different competitiveness and response strategies to water and nitrogen resources. Annuals such as *Tribulus terretris* and *Salsolacollina* can quickly accumulate resources and complete life cycles in a short period of time under grazing and drought conditions. *Chenopodium acuminatum*, *Tribulus terretris* and *Pennisetum centrasiaticum* also use water, nitrogen and other resources efficiently to accelerate their growth.

Coverage is an important index of vegetation growth status and trend [44], and species height, richness and productivity are important parameters of plant community structure and function [45]. Our study showed that grazing had a significantly negative effect on CWM_height_, which was consistent with the results of long-term grazing leading to the apparent miniaturization of plants, an avoidance strategy of grassland plants [46]. Grazing significantly reduced litter, due to the fact that during the non-growing season grazing process, livestock feed directly on surface litter when winter and spring pasture supplies are insufficient [47]. In addition, the feeding, stampede and excretion behavior of herbivores leads to an increase in the number and activity of microorganisms in the soil, which will accelerate the rate of litter decomposition [48]. Previous studies believed that grazing affected the plant density of dominant species and created various habitats in the plant community [49]. Consistent with previous studies, grazing increased plant density in our study because grazing greatly increased the community gap and plant regeneration rate, providing a broader space for plant growth [50,51]. In this study, grazing did not have a significant effect on productivity reduction, and the management measures of grazing in the non-growing season and grazing prohibition in the growing season promoted the compensatory growth of grassland plants, which was conducive to the efficient management and utilization of grassland.

Previous studies have shown that rainfall changes can significantly affect species richness, Simpson dominance index and the Shannon–Wiener diversity index [52] but that they have no significant effect on Pielou evenness index [53]. Experiments simulating extreme drought have found that drought can reduce plant height, coverage and diversity, and ultimately reduce productivity [54]. Consistent with the results of previous studies, the CWM_height_ and coverage of −60% and −60 d in our study were reduced, and −60% significantly reduced the Shannon–Wiener diversity index and increased Simpson dominance index. The main reason is that the drought leads to the continuous extinction of non-drought-tolerant species, but also makes the drought-tolerant plants survive, and the species diversity index decreased while the dominance index increased, and finally, there tends to be a simplification of the plants in the community.

The relationship between plant density and soil moisture is mutually beneficial and antagonistic [55]. In our study, increasing rainfall increased plant density, mainly because increased rainfall would result in an increase in the number of individual plants per unit area in time to improve water use efficiency. In addition, the −60 d treatment also increased plant density, mainly because the plant community reduced surface evaporation by increasing plant density and ultimately improved soil water use efficiency when there was no rainfall in the early growing season.

In our study, the −60 d treatment reduced AGB, while the −60% treatment did not have a significant effect on AGB, which was inconsistent with the previous research results on the reduction of biomass during the growing season [56]. The main reason for this is that the rainfall distribution in 2018 was relatively uniform, and the residual rainfall after −60% of the treatment can also meet the water required for plant growth [41], while −60 d affects the water required by plants in the early stage of growth, and ultimately affects plant growth and reduces biomass. There were large differences in the results of studies on the response of belowground biomass to rainfall changes, which showed that belowground biomass would increase, decrease and remain unchanged as rainfall increased [57,58,59]. In our study, the rainfall changes had no significant effect on the BGB of the 0–10 cm soil layer, which was mainly because the overall rainfall in the Horqin area was small, the soil texture was loose and the water holding capacity was weak, so increasing rainfall or decreasing rainfall would not affect the accumulation of soil surface biomass. However, the −60% treatment increased the BGB at soil depths of 10–20 cm, 20–40 cm and 40–60 cm, indicating that in the Horqin sandy grassland, deeper BGB would be affected by extreme drought. This is consistent with the optimal allocation hypothesis, that is, as rainfall decreases, plants will allocate more biomass to the subsurface part and eventually lead to an increased root-to-shoot ratio [60].

Most studies have shown that nitrogen deposition affects the composition, structure and function of grassland plants [61]. Our study showed that nitrogen addition reduced species richness and the Shannon–Wiener diversity index and increased Simpson dominance index, which is consistent with the findings of most studies that nutrient addition led to decreased species diversity [62]. On the one hand, the increase in available nitrogen in the soil leads to the competition for nutrients in the underground part of the plant and to the competition for light in the aboveground part, and the shade-intolerant and shorter plants are excluded by competitive plants, resulting in a decline in species richness [51]. On the other hand, nitrogen addition improves the soil nutrient status, and the increase in plant individuals also leads to a decrease in plant density, resulting in the loss of rare species in the community, thereby reducing species richness [63]. In addition, nitrogen fertilization also significantly increased AGB, which is consistent with the findings that nitrogen can effectively improve the productivity of degraded grasslands [64].

Previous studies in the Horqin Sandy Land have shown that water and nitrogen interaction can significantly affect the composition, structure and productivity of the grassland community [65]. In our study, the interaction between rainfall and nitrogen had no significant effect on grassland vegetation characteristics or soil water and nitrogen content. On the one hand, the reason is that the evenly distributed rainfall in the experiment year alleviates the inhibitory effect of nitrogen on plant growth [41]. Another reason is that in semiarid regions, vegetation growth is more sensitive to moisture conditions, and it takes longer to respond to nitrogen [66].

The structural equation model of our study showed that rainfall changes could increase CWM_height_ and thus promote AGB. Soil nitrogen content also promoted AGB, indicating that the structure and function of plant communities and their stability in this area are directly or indirectly affected by increased rainfall and nitrogen deposition. Grazing and rainfall could also indirectly affect AGB by influencing CWM_height_, indicating that plant height is the main indicator affecting plant growth rate and plant tissue quality structure support, and it is also the main factor affecting productivity. Grazing had an indirect effect on BGB by increasing plant density and reducing soil nitrogen. In general, the vegetation characteristics, soil water and nitrogen content changes and their interrelationships in the Horqin sandy grassland were the results of the joint action of a variety of environmental factors. In addition to the grazing, rainfall changes and nitrogen addition factors, the influence of topographic factors and human activity interference cannot be ignored. Future research on the influencing factors of plant community structure and function in this region needs to be further strengthened.

## 5. Conclusions

Grazing, rainfall changes and nitrogen addition had a strong impact on the composition, structure and function of the vegetation community in the Horqin sandy grassland. The main manifestation is that grazing reduced CWM_height_ and litter, and increased plant density. +60% increased CWM_height_; −60% and −60 d decreased CWM_height_ and coverage; −60 d, +60% and +60 d increased plant density; −60% increased the Simpson dominance index and decreased the Shannon–Wiener diversity index; −60 d decreased the AGB; −60% increased BGB at soil depths of 10–20 cm, 20–40 cm and 40–60 cm. Nitrogen addition reduced species richness and the Shannon–Wiener diversity index, and increased the Simpson dominance index and AGB. Rainfall changes and soil nitrogen directly affected AGB, grazing and rainfall changes indirectly affected AGB by inducing changes in CWM_height_, and grazing can indirectly affect BGB by affecting plant density and soil nitrogen. Therefore, grazing, rainfall changes and nitrogen addition not only affected the vegetation characteristics of sandy grasslands, but also affected the relationship between soil water and nitrogen content and vegetation characteristics, and ultimately changed the species composition, structure and function of sandy grasslands. Grazing in the non-growing season and grazing prohibition in growing season promoted the compensatory growth of grassland plants. Our study is conducive to the efficient management and sustainable development of grasslands.

## Figures and Tables

**Figure 1 ijerph-20-00960-f001:**
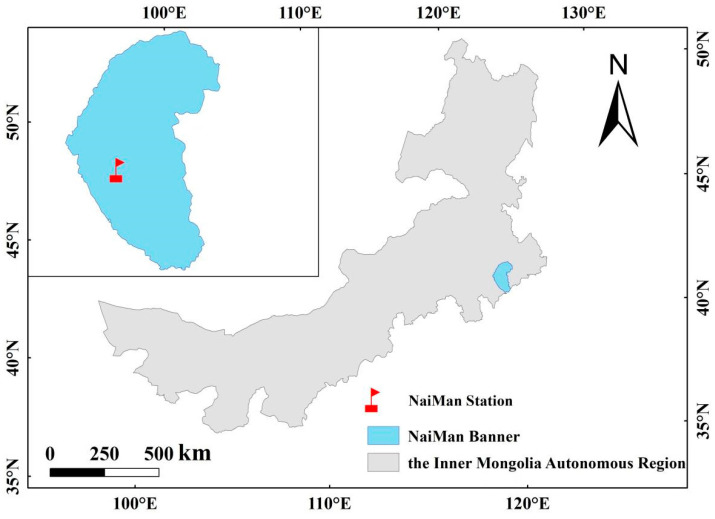
Location of the study area.

**Figure 2 ijerph-20-00960-f002:**
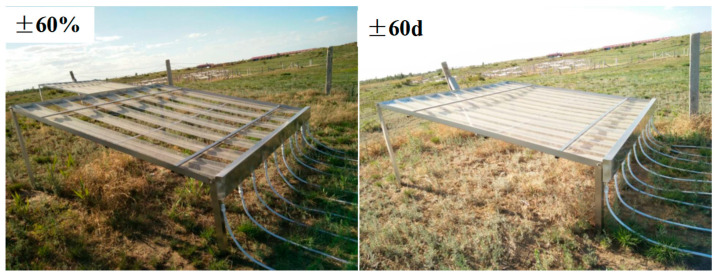
The rainfall-controlled devices used to increase and decrease rainfall.

**Figure 3 ijerph-20-00960-f003:**
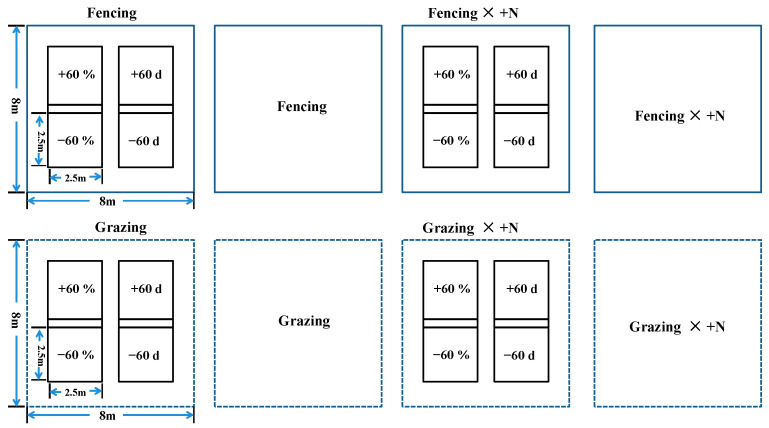
Layout of the test plots.

**Figure 4 ijerph-20-00960-f004:**
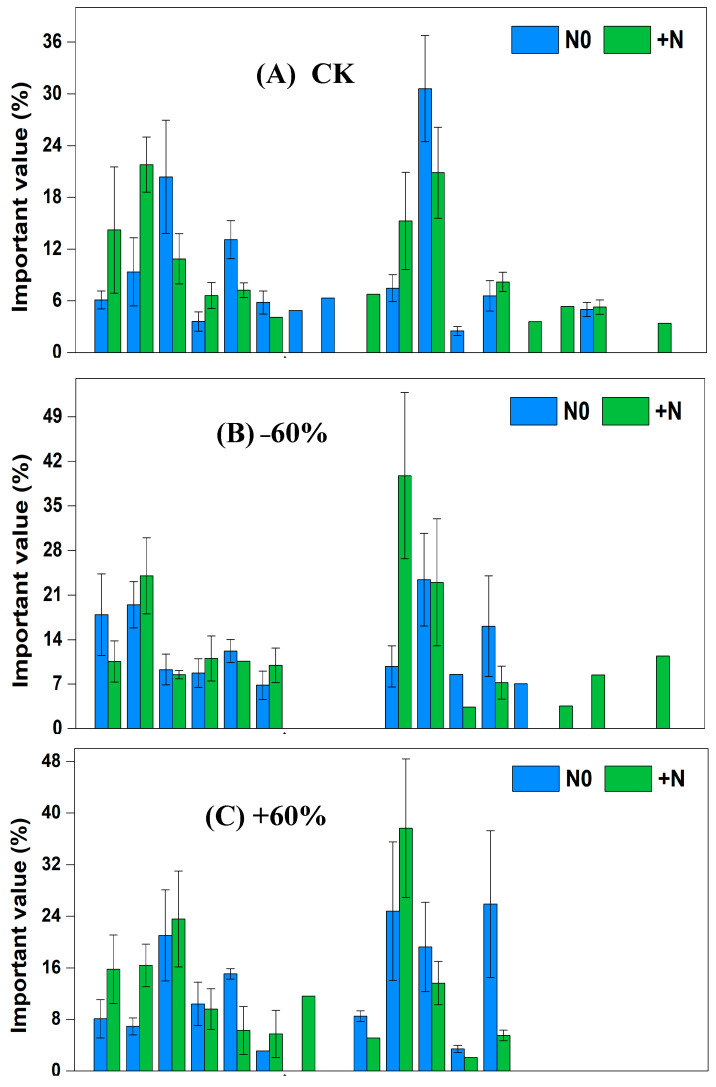
Changes of important values of vegetation in the sandy grassland under rainfall changes and nitrogen addition under fencing. Note: N0 is no nitrogen; +N is nitrogen addition.

**Figure 5 ijerph-20-00960-f005:**
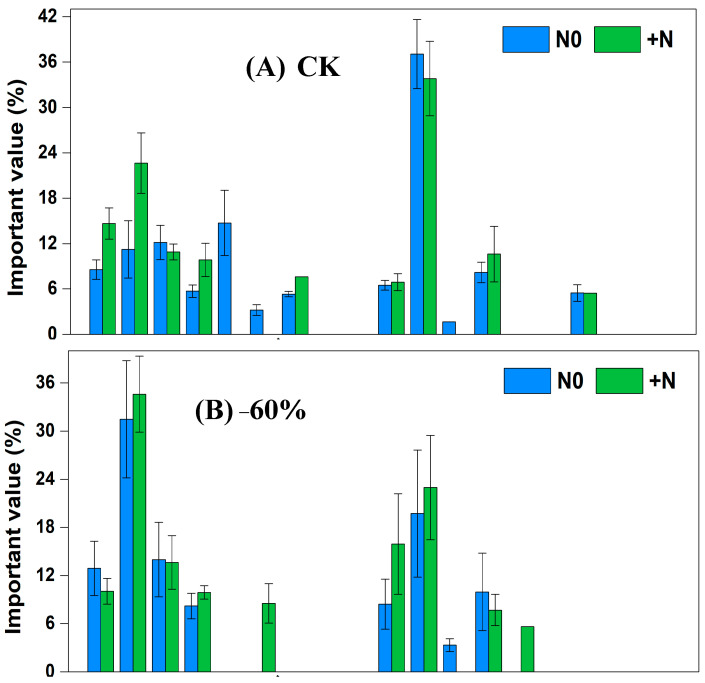
Changes of important values of vegetation in the sandy grassland under rainfall changes and nitrogen addition under grazing.

**Figure 6 ijerph-20-00960-f006:**
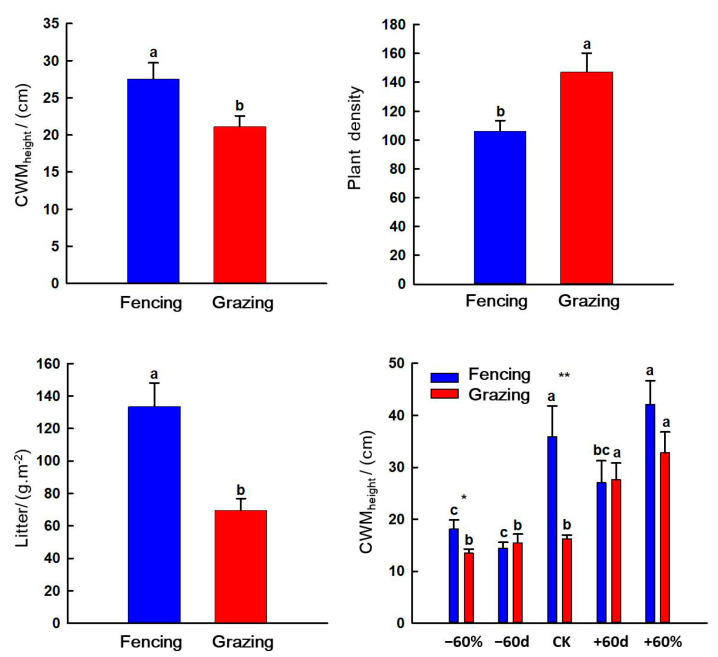
Effects of grazing on vegetation characteristics in the sandy grassland. Note: Different little letters indicate significant differences among grazing and fencing (*p* < 0.05). * indicates significant differences between grazing and fencing treatment under specific rainfall treatments (*p* < 0.05); ** indicates extremely significant differences between grazing and fencing treatment under specific rainfall treatments (*p* < 0.01).

**Figure 7 ijerph-20-00960-f007:**
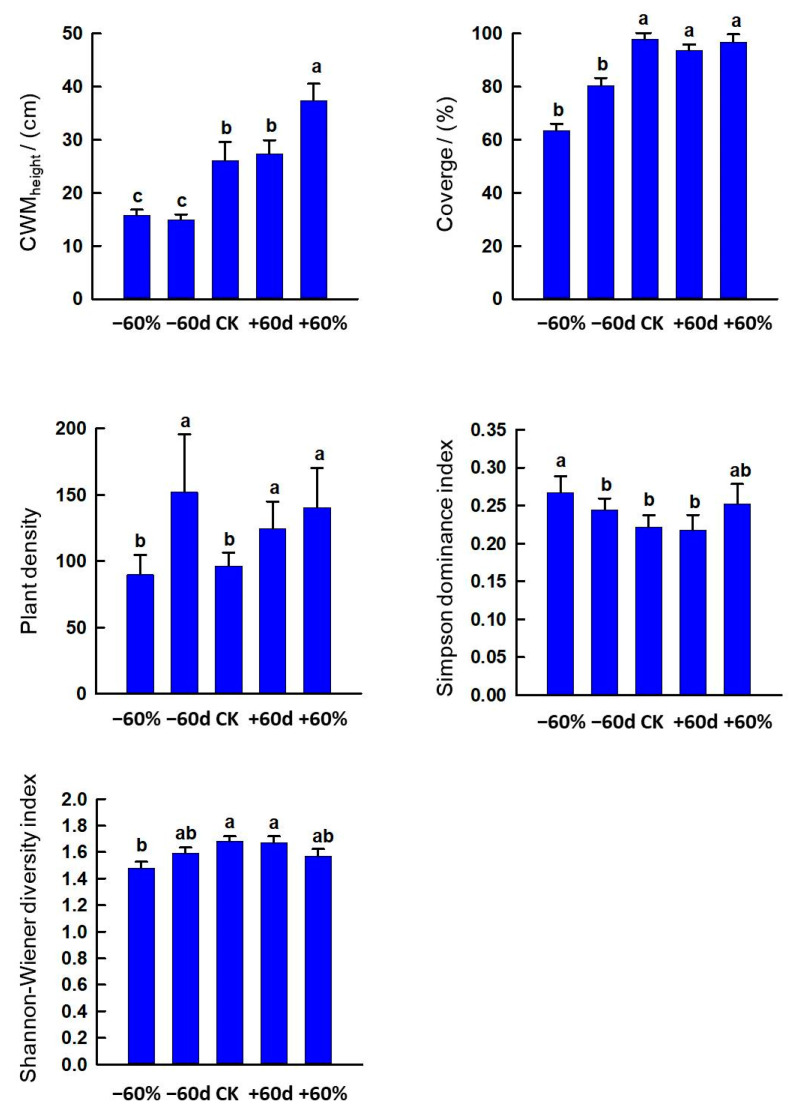
Effects of rainfall changes on vegetation characteristics in the sandy grassland. Note: Different little letters indicate significant differences among different rainfall treatments (*p* < 0.05).

**Figure 8 ijerph-20-00960-f008:**
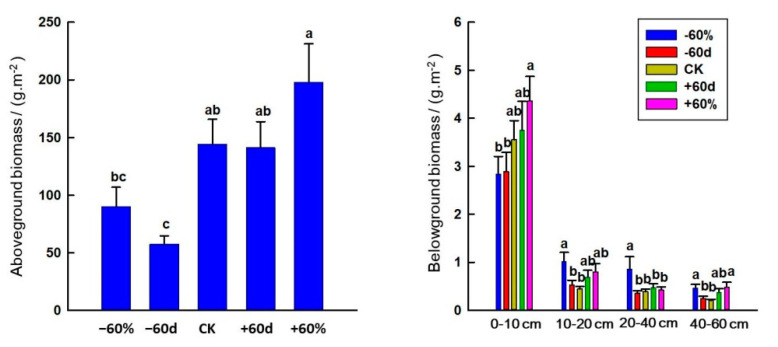
Effects of rainfall changes and soil depth on aboveground and belowground biomass in the sandy grassland. Note: Different little letters indicate significant differences in aboveground biomass among different rainfall treatments (*p* < 0.05), as well as significant differences in belowground biomass among different rainfall treatments in the same soil layer (*p* < 0.05).

**Figure 9 ijerph-20-00960-f009:**
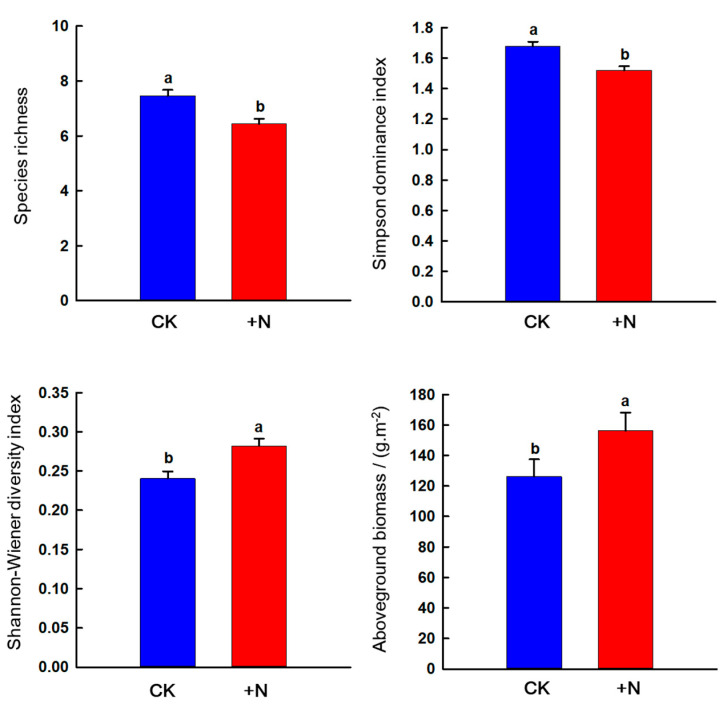
Effects of nitrogen addition on vegetation characteristics and aboveground biomass in the sandy grassland. Note: Different little letters indicate significant differences in nitrogen treatment (*p* < 0.05).

**Figure 10 ijerph-20-00960-f010:**
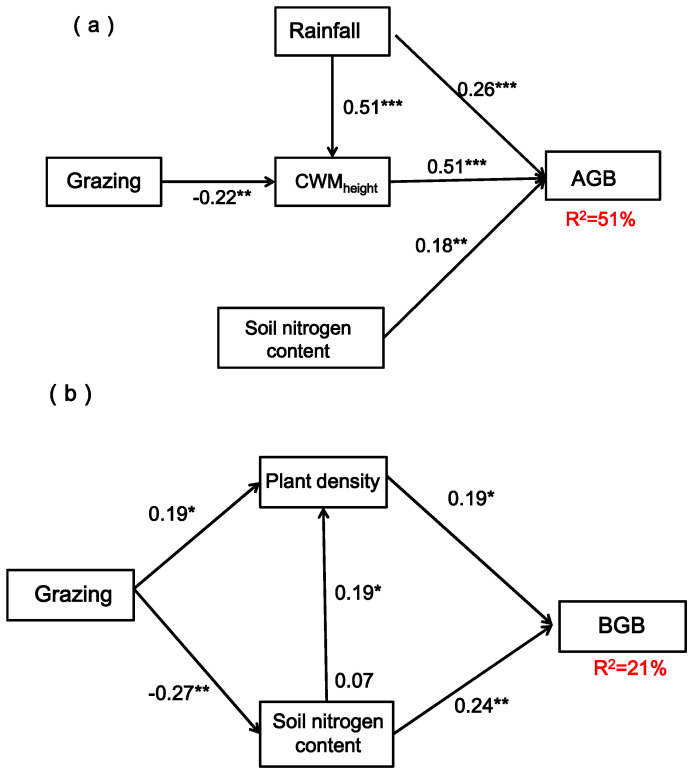
Structural equation model showing all interaction pathways of AGB (aboveground biomass) and BGB (belowground biomass) of grazing, rainfall, soil nitrogen content, plant community height (CWM_height_) and plant density. Subfigure (**a**) shows the structural equation model affecting aboveground biomass (AGB); subfigure (**b**) shows the structural equation model affecting belowground biomass (BGB). Note: The single-headed arrows represent paths in this conceptual model; the arrow width is proportional to the strength of the relationship. Standardized regression weights (along the path) and total variance are explained as a result of all predictors pointing to that variable. *, ** and *** indicate statistically significant paths at *p* < 0.05, *p* < 0.01 and *p* < 0.001, respectively.

**Table 1 ijerph-20-00960-t001:** Total rainfall and monthly rainfall during the 2018 growing season.

Increasing Rainfall or Decreasing Rainfall Treatment	CK	−60%	60%	−60 d	+60 d
Total rainfall (mm)	376.36	150.55	602.18	270.30	482.44
Rainfall in May (mm)	35.17	14.07	56.27	0.00	70.34
Rainfall in June (mm)	70.91	28.36	113.45	0.00	141.82
Rainfall in July (mm)	108.19	43.28	173.11	108.19	108.19
Rainfall in August (mm)	162.09	64.84	259.35	162.09	162.09

Note: CK is actual rainfall; −60% is decreasing actual rainfall by 60% from May to August; +60% is increasing actual rainfall by 60% from May to August; −60 d is decreasing actual rainfall by 100% from May to June; +60 d is increasing actual rainfall by 100% from May to June; the same below.

**Table 2 ijerph-20-00960-t002:** Variance analysis of the response of vegetation characteristics to grazing, rainfall and nitrogen addition in the sandy grassland.

Treatment	CWM_height_/ (cm)	Coverage/(%)	Species Richness	Plant Density	Litter Biomass/ (g·m^−2^)	D Index	H Index	J Index	AGB/(g·m^−2^)	BGB/(g·m^−2^)
Grazing	9.47 **	0.10	0.13	8.20 **	14.94 **	5.25	3.43	4.64	1.49	0.21
Rainfall	15.76 **	48.76 **	1.93	3.30 *	0.54	4.59 **	3.79 **	2.83	18.612 **	1.13
nitrogen	0.52	1.77	13.89 **	0.68	0.27	12.30 **	17.61 **	1.85	5.08 *	0.05
grazing×rainfall	3.33 *	0.74	1.19	1.09	0.38	1.46	1.49	0.82	0.49	0.20
grazing×nitrogen	0.38	2.02	3.25	1.86	0.52	0.03	1.28	0.66	0.01	1.06
rainfall × nitrogen	0.58	1.26	0.87	1.87	1.30	0.63	0.63	0.39	1.75	1.11
grazing × rainfall × nitrogen	1.59	1.26	0.70	0.79	1.31	2.33	1.97	1.15	0.18	0.56

Note: D index is Simpson dominance index; H index is the Shannon–Wiener diversity index; J index is Pielou evenness index; AGB is aboveground biomass; BGB is belowground biomass; the same below. * and ** indicate statistically significant levels at *p* < 0.05 and *p* < 0.01.

## Data Availability

Not applicable.

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
