# Peer review of "Effects of Grazing, Extreme Drought, Extreme Rainfall and Nitrogen Addition on Vegetation Characteristics and Productivity of Semiarid Grassland"

_ijerph, 2023, doi:10.3390/ijerph20020960_

Round 1

Reviewer 1 Report

Review of manuscript "Effects of grazing, extreme drought, extreme rainfall and nitrogen addition on vegetation characteristics and productivity of semi-arid grassland" (ijerph-1993604)

Dear authors, your research tries to analyze the influences of overgrazing, extreme climate and nitrogen addition on vegetation characteristics in the semi-arid grassland of China. It is a complete manuscript and fits the aims and scope of the journal topic. Nevertheless, the authors need to highlight the novelty of their research building upon previous research. Therefore, "Major Revision" is required to largely improve this manuscript. Specifically, the reviewer has the following comments and suggestions:

(1) The Abstract and Introduction Section: both of these two parts are weak because the authors did not highlight the purpose, originality, and novelty of this study from an international perspective. As a consequence, reviewers cannot figure out why this research must be performed in this context. Some studies have already analyzed the influences of overgrazing, extreme climate and nitrogen addition on vegetation characteristics in some other countries and regions.

(2) A criticism-featured and thorough Section of Literature Review must be provided. The literature review in the current manuscript is very insufficient.

(3) Figure 1: in this figure, what is the "Naiman Station"? Did the authors mean that your study was just performed in a small station rather than the whole Banner?

(4) Section 2.2 Experimental design and measuring: in this part, the authors did not provide the specific details about the experimental samples. For example, what are the detailed locations, orientations, etc.? These kinds of information will affect the performance of the experiments.

(5) Section 2.4 Methods for measuring species diversity: in this part, the definition of the "species" should be explained. I guess only the vegetation species have been taken into account? What about the diversity of insects or animals?

(6) What is the definition of the "plant community height"? Does this mean that the canopy height?

(7) Section 3.1 Rainfall characteristics of growing season in the study area: in this part, only the rainfall data in 2018 were taken into account in this experiment, which may not be enough for representing the rainfall characteristics in this study area. The rainfall data in one specific year may be subject to uncertainty and variability.

(8) The tables shown in the Section 3.2 and Section 3.3 should be better clearly presented in several figures. The data in those tables are hardly to be distinguished.

(9) Takeaway for practice is also encouraged to be included in this manuscript. It should be clear enough to present your policy recommendations for both local and international practice.

(10) Section 4 Discussions: in this part, the authors should also provide limitations and recommendation for improvement in the discussion section.

(11) Section 5 Conclusions: in this part, what is new in this study? The novelty of this research should be apparent in the manuscript. How to link the findings and conclusions in this paper with the previous findings and conclusions?

Author Response

Dear reviewer:

I have made comprehensive revisions to the content of the manuscript according to the reviewers' comments, in addition, I myself have found the shortcomings of the manuscript in the revision process and made the following three changes:

  1. 1. Adjust the color of the picture to make the picture clearer;
  2. The rainfall in 2018 was checked, and the previous total rainfall was found to be biased, and the rainfall was recalculated in the revised manuscript and modified;
  3. The abstract and introduction have been considerably improved.
  4. Tables 5 and 6 serve as supporting evidence in the article and are therefore presented as a Supplementary.

The following section is my response to the reviewers' comments, the modified parts of the full manuscript are covered with grayscale.

Review of manuscript "Effects of grazing, extreme drought, extreme rainfall and nitrogen addition on vegetation characteristics and productivity of semi-arid grassland" (ijerph-1993604)

Dear authors, your research tries to analyze the influences of overgrazing, extreme climate and nitrogen addition on vegetation characteristics in the semi-arid grassland of China. It is a complete manuscript and fits the aims and scope of the journal topic. Nevertheless, the authors need to highlight the novelty of their research building upon previous research. Therefore, "Major Revision" is required to largely improve this manuscript. Specifically, the reviewer has the following comments and suggestions:

  • The Abstract and Introduction Section: both of these two parts are weak because the authors did not highlight the purpose, originality, and novelty of this study from an international perspective. As a consequence, reviewers cannot figure out why this research must be performed in this context. Some studies have already analyzed the influences of overgrazing, extreme climate and nitrogen addition on vegetation characteristics in some other countries and regions.

Reply:The introduction department has been systematically sorted out according to the comments of the reviewers, and it has been rewritten on the basis of the previous one.

  • A criticism-featured and thorough Section of Literature Review must be provided. The literature review in the current manuscript is very insufficient.

Reply:I have made revisions based on the comments of the reviewers.

  • Figure 1: in this figure, what is the "Naiman Station"? Did the authors mean that your study was just performed in a small station rather than the whole Banner?

Reply:Yes, the reviewer is right. The Naiman Desertification Research Station is a scientific research station set up in Naiman Banner, and our experiments are based on the Naiman Desertification Research Station. We selected a sandy grassland in the Naiman Banner area for grazing, rainfall change and nitrogen addition experiments, which is a typical representative grassland distributed in the Naiman Banner area.

  • Section 2.2 Experimental design and measuring: in this part, the authors did not provide the specific details about the experimental samples. For example, what are the detailed locations, orientations, etc.? These kinds of information will affect the performance of the experiments.

Reply:Additional clarifications have been made in accordance with the comments of the reviewers: This study was conducted in 2018 and selected as the re-search object the sandy grassland  that began fencing grazing at the Naiman Desertification Research Station in 2015 (120°42′E,42°55′N).

  • Section 2.4 Methods for measuring species diversity: in this part, the definition of the "species" should be explained. I guess only the vegetation species have been taken into account? What about the diversity of insects or animals?

Reply:Species diversity has been changed to plant species diversity.

  • What is the definition of the "plant community height"? Does this mean that the canopy height?

Reply:The definition of plant community height is not canopy height, because the experimental plot is a sandy grassland dominated by annual and perennial grasses, there are no similar plant species such as trees, and the dominant species in the grassland are described in Part 2.1, and the dominance of plant species is also described in detail in Parts3.1 and 3.2.

  • Section 3.1 Rainfall characteristics of growing season in the study area: in this part, only the rainfall data in 2018 were taken into account in this experiment, which may not be enough for representing the rainfall characteristics in this study area. The rainfall data in one specific year may be subject to uncertainty and variability.

Reply:The average rainfall in the 20-year growing season (May-August) in this region is 380.03 mm, and the rainfall in the 2018 (May-August) growing season is 376 mm, so the rainfall in this year is neither a dry year nor a wet year, which is typical. In addition, our experiments were stabilized after previous years before continuing to do experiments, so the results of 2018 are reliable and have certain scientific significance. Future results will be needed to demonstrate this achievement.

(8) The tables shown in the Section 3.2 and Section 3.3 should be better clearly presented in several figures. The data in those tables are hardly to be distinguished.

Reply:It has been revised in accordance with the comments of the reviewers.

(9) Take away for practice is also encouraged to be included in this manuscript. It should be clear enough to present your policy recommendations for both local and international practice.

Reply:Amendments have been made in the summary, discussion and conclusion sections.

(10) Section 4 Discussions: in this part, the authors should also provide limitations and recommendation for improvement in the discussion section.

Reply:It has been revised in accordance with the comments of the reviewers.

(11) Section 5 Conclusions: in this part, what is new in this study? The novelty of this research should be apparent in the manuscript. How to link the findings and conclusions in this paper with the previous findings and conclusions?

Reply:It has been revised in accordance with the comments of the reviewers.

Reviewer 2 Report

Dear authors,

Your complex study is interesting but some applied approaches are questionable, especially in data analysis. Below are, firstly, the most sufficient questions and objections.

Figure 2, Table 1: in the variants “+60%” and “+60d”, regulated increased rainfall is in precise given ratio with natural rainfall value in the experiment year (2018), not with long-term observational annual value that is referred in lines 119-120. Please, explain how does the device guarantee such precise correspondence between regulated regime and rainfall of current season? (I can imagine it is possible to regulate decreased rainfall precisely enough: for example, by size of the device gaps; but I don’t how precise quantity of additional rainfall can be guaranteed.)

Figure 2: does the device decrease light availability for grass? Moreover, does it decrease grass and soil heating (that can lead to decreased transpiration and evaporation)? I see, these things are not discussed in Section 4 that leads to a primitive picture of vegetation processes.

Lines 165-166: what is “species height”? Is it average value amongst all species individuals, or their mode value or what? What is “total height”? Is it average height of all plants in community?

Lines 338-339: “Models based on grazing, rainfall, soil nitrogen and CWMheight provided the best explanation for AGB …, explaining 51% of AGB in total (Figure 8a).” Biomass is used in calculation of CWMheight (see your formula 5), therefore modelling AGB through CWMheight is a cycled calculation. The model would be correct if CWMheight was excluded.

Lines 320-321: “CWMheight was extremely significantly positively correlated with coverage and AGB (P< 0.01).” Biomass is used in calculation of CWMheight (see your formula 5), so correlation between CWMheight and AGB is a trivial result which is not worth to mention.

Lines 323-325: the same note on correlation between species richness and Shannon-Weaver index because the first one impacts on calculation of the second one (by number of the sum members: see your formula 3).

Notes on structure and style of the manuscript:

Lines 47-50: Here, unclear is the discourse transition from climate to nitrogen deposition. Whether there are direct relations between climate (global warming) and nitrogen deposition or not? Maybe, the phrases in lines 64-68 would be suitable here as a logical connection.

Lines 60-61: “surface biological crusts”. Which crusts do you mean? Are they algal, fungal, lichen ones or somewhat else? It is worth to indicate this more specifically.

Lines 123-125: information on the device producer should be added as well as references on other studies that deal with such devices. Particularly, your study [49] should be mentioned here, and other examples are desirable.

Figure 3: why the variant “-60%” is repeated twice in each of the four plots marked up? Where is the variant “-60d”?

Figure 3 vs. sections 3.2, 3.3, Tables 2 and 3: the symbol N0 should be shown in Figure 3 with appropriate explanation. Table 2 should be referred in the first phrase of section 3.2, the same should be for Table 3 and section3.3.

Lines 145-146: it seems reasonable to collect soil and vegetation characteristic samples from each of 48 selected quadrates. Did random layout guarantee that characteristic data were obtained from all study plots?

Lines 150-152: what a way or tool was used to detect total vegetation cover and sub-coverage of each species?

Line 163: usually in articles, section titles do not include references. The publication [35] should be referred within the text of section 2.4.

Lines 175, 176, 178-179: unclear is summation from species i to total number of plant individuals in community. Please, check correctness of the formulas (2) and (3). It seems that either S is a number of species (not of plant individuals) in community or the final member of the sum must be numbered with another letter (not S).

Lines 181-183: “sum of the heights of all plants in the community and their relative biomass, which is the mean value of the heights in the community”. Firstly, according to formula (5), CWMheight is sum of products (each product is biomass multiplied by height). Secondly, why biomass is the mean value of heights? What do you imply?

Table 1: in experiment reports, such data are usually placed in the Methods section (in this article, suitable place is near lines 121-123) because they represent the experiment input conditions.

Section 3.2: the text is a simple retelling of Table 2 content. Instead, mentioned should be the most pronounced effects (for each treatment, which is especially notable species’ response), whereas theoretically expected results should be missed. Therefore, the text should be strongly reduced, and Table 2 should be referred in the first phrase of section 3.2.

Section 3.3 and Table 3: the same notes as for section 3.2. Section 3.3 should be reduced, and in its first phrase Table 3 should be referred. Perhaps, integration of shortened sections 3.2 and 3.3 would be suitable.

Table 4: in notations, explained should be what do mean numbers in table cells and which cells are marked with one asterisk or two ones.

Figure 4: on bottom right diagram, the applied two colors should be more contrast between each other.

Figures 4, 5, 6, 7: in caps, explained should be what whiskers denote.

Figures 5, 6, 7: in caps, explained should be whether the presented data demonstrate grazing treatment or fencing treatment, or integrated results from the two ones.

Line 333: “AGB is positively correlated with soil water content (P< 0.05) (Table5).” Where is AGB in Table 5??? The same note applies to AGB mentions in lines 321, 323, 331.

Lines 334, 348: in table titles, which number 5 is correct? Which table is implied in lines 321,323, 331, 333?

Table 5 after lines 348-349: in notations, explained should be which table cells are marked with one or more asterisks.

Lines 327-328: duplicated phrases “The Simpson dominance index is positively correlated with litter (P< 0.05)” and “Litter is significantly positively correlated with Simpson dominance index (P< 0.05)”

Lines 321-322: “Coverage is negatively correlated with Simpson dominance index (P< 0.05), significantly positively correlated with Pielou evenness index (P< 0.05), …”. These correlations (although significant) seem too weak to mention.

The same note on lines 325-326: “Species density was extremely positively correlated with soil nitrogen content (P< 0.01).” By the way, in table, this parameter is called “Plant density” (the same note applies to lines 382, 398, 399, 401, 403, 462, 464).

The same note (weak correlation) on lines 327-329: “Litter is … extremely significantly negatively correlated with Pielou evenness index (P< 0.01).”

Line 333: the same note on correlation between AGB and soil water content.

Lines 329-331: “Simpson dominance index is … negatively correlated with AGB (P< 0.05).” This correlation (although significant) is too weak to discuss it seriously.

Figure 8 (a, b): in cap, explained should be which arrows are marked with one or more asterisks and what does mean “Err”

Lines 391-393: the phrase duplicates the previous one (lines 388-390).

Besides, some notes on grammar:

Lines 110-112: errors in plant Latin names. Should be: Tribulus terrestris L, Setaria viridis (L.) Beauv [with space between genus and species names], and Pennisetum centrasiaticum Tzvel [italic font is necessary for genus and species names]. The same should be checked in lines 214, 216, 221, 224, 226, 227, 231, 233, 236, and further through the text, including lines 354-368.

Line 360, 364: “Salsolacollina”, “Tribulusterrestris”. Inserted should be a space between genus and species names.

Lines 354-368: Latin names should be in italics

Line 147: should be written “Vegetation characteristic indicators included plant community height, …”

Line 273: “Coverge” (should be “Coverage”)

Lines 355, 371, 382, 389, 399, 424, 439 and others of similar context: “our study” would be more clear than “this study”

Author Response

Dear reviewer:

I have made comprehensive revisions to the content of the manuscript according to the reviewers' comments, in addition, I myself have found the shortcomings of the manuscript in the revision process and made the following three changes:

  1. 1. Adjust the color of the picture to make the picture clearer;
  2. The rainfall in 2018 was checked, and the previous total rainfall was found to be biased, and the rainfall was recalculated in the revised manuscript and modified;
  3. The abstract and introduction have been considerably improved.
  4. Tables 5 and 6 serve as supporting evidence in the article and are therefore presented as a Supplementary.

The following section is my response to the reviewers' comments, the modified parts of the full manuscript are covered with grayscale.

Reviewer 2

Your complex study is interesting but some applied approaches are questionable, especially in data analysis. Below are, firstly, the most sufficient questions and objections.

  1. Figure 2, Table 1: in the variants “+60%” and “+60d”, regulated increased rainfall is in precise given ratio with natural rainfall value in the experiment year (2018), not with long-term observational annual value that is referred in lines 119-120. Please, explain how does the device guarantee such precise correspondence between regulated regime and rainfall of current season? (I can imagine it is possible to regulate decreased rainfall precisely enough: for example, by size of the device gaps; but I don’t how precise quantity of additional rainfall can be guaranteed.)

Reply:First of all, we fully consider the wind direction in the process of installing the increasing and decreasing rainfall device, in addition, in the case of a large enough area, the wind will cause a small error, this small effect can be ignored, there are many articles on the use of the device for testing has also been published, so the device is generally recognized and reasonable.

[1]Sun SS, Liu XP, Zhao XY et al. Annual Herbaceous Plants Exhibit Altered Morphological Traits in Response to Altered Precipitation and Drought Patterns in Semiarid Sandy Grassland, Northern China. Frontiers in Plant Science, 2022, 13: 756950.

[2]Hu Y, Li XY, Guo AX et al. Species diversity is a strong predictor of ecosystem multifunctionality under altered precipitation in desert steppes. Ecological Indicators, 2022, 137:108762.

[3] Zhang J, Zuo X, Zhao X, et al. Effects of rainfall manipulation and nitrogen addition on plant biomass allocation in a semiarid sandy grassland. Scientific Reports, 2020,10,1:9026.

  1. Figure 2: does the device decrease light availability for grass? Moreover, does it decrease grass and soil heating (that can lead to decreased transpiration and evaporation)? I see, these things are not discussed in Section 4 that leads to a primitive picture of vegetation processes.

Reply:The rain baffle of the device is made of polyester plastic with good light transmission, which can ensure the normal photosynthesis of plants, and the distance between the device and the ground is 1.5 meters, which has enough space to ensure the transpiration of plants.

  1. Lines 165-166: what is “species height”? Is it average value amongst all species individuals, or their mode value or what? What is “total height”? Is it average height of all plants in community?

Reply:It has been revised in accordance with the comments of the reviewers. In the above formula, the relative height is the height of a particular plant divided by the sum of the heights of all plants in the community, the relative coverage is the coverage of a plant divided by the sum of the coverage of all plants in the community, and the relative biomass is the biomass of a particular plant divided by the sum of the biomass of all plants in the community.

  1. Lines 338-339: “Models based on grazing, rainfall, soil nitrogen and CWMheight provided the best explanation for AGB …, explaining 51% of AGB in total (Figure 8a).” Biomass is used in calculation of CWMheight (see your formula 5), therefore modelling AGB through CWMheight is a cycled calculation. The model would be correct if CWMheight was excluded.

Reply:Coverage and biomass are generally used to calculate the weights, and the optimal indicator is biomass. The biomass used to calculate the relative weight is the proportion of each plant’s abundance. The dominance of each species in the actual community is different, and only after the weight and trait values of each species are multiplied and weighted can the characteristics of the community be truly reflected. I understand that reviewers are questioning that my model may have collinearity interpretation problem. Collinearity can only be between variables, that is, between height and other traits or soils, AGB is the dependent variable rather than the independent variable, so there is no collinearity problem. I will list a relevant articles to support the validity of my model.

[1]Zuo XA; Zhang J; Lv P et al. Effects of plant functional diversity induced by grazing and soil properties on above- and belowground biomass in a semiarid grassland.

  1. Lines 320-321: “CWMheight was extremely significantly positively correlated with coverage and AGB (P< 0.01).” Biomass is used in calculation of CWMheight (see your formula 5), so correlation between CWMheight and AGB is a trivial result which is not worth to mention.

Reply:According to the reviewer's opinion, in the process of revising the manuscript, I found that the results involved in Table 6 were partially redundant, that is, the correlation analysis in lines 318-333 of 3.5 in the original text was only a statement of the results, which did not have enough support for the results of the study, so this part was deleted.

  1. Lines 323-325: the same note on correlation between species richness and Shannon-Weaver index because the first one impacts on calculation of the second one (by number of the sum members: see your formula 3).

Reply:This part has been deleted in the revised manuscript.

Notes on structure and style of the manuscript:

  1. Lines 47-50: Here, unclear is the discourse transition from climate to nitrogen deposition. Whether there are direct relations between climate (global warming) and nitrogen deposition or not? Maybe, the phrases in lines 64-68 would be suitable here as a logical connection.

Reply:This part has been rewrite in the revised manuscript.

  1. Lines 60-61: “surface biological crusts”. Which crusts do you mean? Are they algal, fungal, lichen ones or somewhat else? It is worth to indicate this more specifically.

Reply:The phrase has been changed to “Grazing leads to the destruction of vegetation canopies and surface biological lichens, increasing the risk of soil erosion”.

  1. Lines 123-125: information on the device producer should be added as well as references on other studies that deal with such devices. Particularly, your study [49] should be mentioned here, and other examples are desirable.

Reply: It has been revised in accordance with the comments of the reviewers.

  1. Figure 3: why the variant “-60%” is repeated twice in each of the four plots marked up? Where is the variant “-60d”?

Reply:This problem is an omission in my drawing, I have corrected it.

  1. Figure 3 vs. sections 3.2, 3.3, Tables 2 and 3: the symbol N0 should be shown in Figure 3 with appropriate explanation. Table 2 should be referred in the first phrase of section 3.2, the same should be for Table 3 and section3.3.

Reply:It has been revised in accordance with the comments of the reviewers.

  1. Lines 145-146: it seems reasonable to collect soil and vegetation characteristic samples from each of 48 selected quadrates. Did random layout guarantee that characteristic data were obtained from all study plots?

Reply:In order to ensure the rationality of the data, the different treatments in our 48 plots are randomly set, before investigating the vegetation characteristics, we will visually check the plant species in the sample, select the unit plot containing the most dominant species in the sample for vegetation investigation, and each treatment also has 6 replicates, which ensures the credibility of the data.

  1. Lines 150-152: what a way or tool was used to detect total vegetation cover and sub-coverage of each species?

Reply:It has been modified in the original text to read as follows: First, the total coverage of the vegetation within the sample is estimated by visual inspection, secondly, the sub-coverage of each species is estimated by visual inspection.

  1. Line 163: usually in articles, section titles do not include references. The publication [35] should be referred within the text of section 2.4.

Reply:I have made changes based on the reviewer's comments.

  1. Lines 175, 176, 178-179: unclear is summation from species i to total number of plant individuals in community. Please, check correctness of the formulas (2) and (3). It seems that either S is a number of species (not of plant individuals) in community or the final member of the sum must be numbered with another letter (not S).

Reply:This problem is indeed an omission and has been corrected as: S is the number of species in the community.

  1. Lines 181-183: “sum of the heights of all plants in the community and their relative biomass, which is the mean value of the heights in the community”. Firstly, according to formula (5), CWMheight is sum of products (each product is biomass multiplied by height). Secondly, why biomass is the mean value of heights? What do you imply?

Reply:Plant community height (CWMheight) is the sum of the heights of all plants in the community and their relative biomass, reflecting the overall height of the community. “which is the mean value of the heights in the community” has a big ambiguity and has been corrected in the original manuscript.

  1. Table 1: in experiment reports, such data are usually placed in the Methods section (in this article, suitable place is near lines 121-123) because they represent the experiment input conditions.

Reply: I have made revisions based on the comments of the reviewers.

  1. Section 3.2: the text is a simple retelling of Table 2 content. Instead, mentioned should be the most pronounced effects (for each treatment, which is especially notable species’ response), whereas theoretically expected results should be missed. Therefore, the text should be strongly reduced, and Table 2 should be referred in the first phrase of section 3.2.

Reply: The table of this section has been expressed in the form of a diagram.

  1. Section 3.3 and Table 3: the same notes as for section 3.2. Section 3.3 should be reduced, and in its first phrase Table 3 should be referred. Perhaps, integration of shortened sections 3.2 and 3.3 would be suitable.

Reply: This section has been modified.

  1. Table 4: in notations, explained should be what do mean numbers in table cells and which cells are marked with one asterisk or two ones.

Reply:This has been explained in detail under the table.

  1. Figure 4: on bottom right diagram, the applied two colors should be more contrast between each other.

Reply: It has been revised in accordance with the comments of the reviewers.

  1. Figures 4, 5, 6, 7: in caps, explained should be what whiskers denote.

Reply:The questions raised by the reviewers have been revised.

Figures 5, 6, 7: in caps, explained should be whether the presented data demonstrate grazing treatment or fencing treatment, or integrated results from the two ones.

Reply:This is a combination of the two treatments. Before exploring the effects of grazing, rainfall change and nitrogen addition on sandy grassland vegetation, we used multivariate ANOVA to analyze the data, the analysis results showed that grazing, rainfall change and nitrogen had different effects on grassland vegetation(Table 2). Therefore, Figure 6-9 specifically showed the influence of a single influencing factor on the vegetation characteristics. Figure 7 indicates the impact of rainfall changes on vegetation characteristics, figure 8 indicated the effect of rainfall changes and soil depth on belowground biomass, and figure 9 indicated the impact of nitrogen addition on vegetation characteristics.

Table 4 Analysis table of variance of vegetation characteristics in sandy grassland

Reply:I have changed to “Variance analysis of the response of vegetation characteristics to grazing, rainfall and nitrogen addition”.

  1. Line 333: “AGB is positively correlated with soil water content (P< 0.05) (Table5).” Where is AGB in Table 5??? The same note applies to AGB mentions in lines 321, 323, 331.

Reply:This section has been modified.

  1. Lines 334, 348: in table titles, which number 5 is correct? Which table is implied in lines 321,323, 331, 333?

Reply:This section has been modified.

  1. Table 5 after lines 348-349: in notations, explained should be which table cells are marked with one or more asterisks.

Reply:This has been explained in accordance with the comments of the reviewers.

  1. Lines 327-328: duplicated phrases “The Simpson dominance index is positively correlated with litter (P< 0.05)” and “Litter is significantly positively correlated with Simpson dominance index (P< 0.05)”

Reply:This section has already been deleted.

  1. Lines 321-322: “Coverage is negatively correlated with Simpson dominance index (P< 0.05), significantly positively correlated with Pielou evenness index (P< 0.05), …”. These correlations (although significant) seem too weak to mention.

Reply:This section has already been deleted.

  1. The same note on lines 325-326: “Species density was extremely positively correlated with soil nitrogen content (P< 0.01).” By the way, in table, this parameter is called “Plant density” (the same note applies to lines 382, 398, 399, 401, 403, 462, 464).

Reply:This section has already been deleted. By reviewing the literature, it was found that the most suitable expression was “plant density”, and it was revised throughout the manuscript.

  1. The same note (weak correlation) on lines 327-329: “Litter is … extremely significantly negatively correlated with Pielou evenness index (P< 0.01).”

Reply:This section has already been deleted.

  1. Line 333: the same note on correlation between AGB and soil water content.

Reply:This section has already been deleted.

  1. Lines 329-331: “Simpson dominance index is … negatively correlated with AGB (P< 0.05).” This correlation (although significant) is too weak to discuss it seriously.

Reply:This section has already been deleted.

  1. Figure 8 (a, b): in cap, explained should be which arrows are marked with one or more asterisks and what does mean “Err”

Reply:The structural equation model was modified and the parameters therein were explained in detail.

  1. Lines 391-393: the phrase duplicates the previous one (lines 388-390).

Reply:The questions raised by the reviewers have been revised.

Besides, some notes on grammar:

  1. Lines 110-112: errors in plant Latin names. Should be: Tribulus terrestris L, Setaria viridis (L.) Beauv [with space between genus and species names], and Pennisetum centrasiaticum Tzvel [italic font is necessary for genus and species names]. The same should be checked in lines 214, 216, 221, 224, 226, 227, 231, 233, 236, and further through the text, including lines 354-368.

Reply:The questions raised by the reviewers have been revised.

  1. Line 360, 364: “Salsolacollina”, “Tribulus terrestris”. Inserted should be a space between genus and species names.

Reply:The questions raised by the reviewers have been revised.

  1. Lines 354-368: Latin names should be in italics

Reply:The questions raised by the reviewers have been revised.

  1. Line 147: should be written “Vegetation characteristic indicators included plant community height, …”

Reply:The questions raised by the reviewers have been revised.

  1. Line 273: “Coverge” (should be “Coverage”)

Reply:The questions raised by the reviewers have been revised.

  1. Lines 355, 371, 382, 389, 399, 424, 439 and others of similar context: “our study” would be more clear than “this study”

Reply:The questions raised by the reviewers have been revised.

Round 2

Reviewer 1 Report

This manuscript is now acceptable for publication.

Author Response

The reviewer said that my paper was ready for publication and there was no further need for revision

Reviewer 2 Report

Dear authors,

I see you did diligently revise the manuscript and really improved it. However, some shortcomings are remained. Please, pay your attention to them.

Lines 188-192: “the relative height is the height of a particular plant divided …” Whether you mean “particular plant” or “particular plant species”? In the first case, what the “particular plant” do you mean? Is it a voluntary selected plant of the species, or what? The same applies to your definitions of relative coverage and relative biomass. Moreover, in the first case one can expect importance of a species to be a sum of values obtained (by formula 1) for all particular plants of this species. Are you sure formula (1) is written correctly? In the second case (if you mean “particular plant species”), how “height of a particular plant species” is calculated? Is it an average height of plants of the species, or what? The same applies to coverage and biomass.

Lines 204-205: I see formula (5) includes the multiplication sign (×) within each term of this sum. That means this formula should be verbally described as “sum of products of the heights of all plants in the community and their relative biomass, …” (In mathematics, a result of multiplication is called “product”). Whereas your phrase “Plant community height (CWMheight) is the sum of the heights of all plants in the community and their relative biomass” implies that height was summed with mass, and this operation is incorrect because of different dimensionality of the two parameters (relative biomass has no measure unit, whilst height do has it).

Line 208: Here, it would be useful to refer explanation to formula (1) about calculation of relative biomass. At this I hope, this explanation will be improved to become clearer, then its current version.

Additional note to style:

Line 51: the word “biological” is unnecessary here. Lichens are always biological. :) Maybe you mean “living surface lichens”? Commonly, simply “surface lichens” will be suitable.

Author Response

Reviewer 2

Dear authors:

I see you did diligently revise the manuscript and really improved it. However, some shortcomings are remained. Please, pay your attention to them.

Lines 188-192: “the relative height is the height of a particular plant divided …” Whether you mean “particular plant” or “particular plant species”? In the first case, what the “particular plant” do you mean? Is it a voluntary selected plant of the species, or what? The same applies to your definitions of relative coverage and relative biomass. Moreover, in the first case one can expect importance of a species to be a sum of values obtained (by formula 1) for all particular plants of this species. Are you sure formula (1) is written correctly? In the second case (if you mean “particular plant species”), how “height of a particular plant species” is calculated? Is it an average height of plants of the species, or what? The same applies to coverage and biomass.

Reply:In the above formula, the relative height is the maximum height of a particular plant species divided by the sum of the maximum heights of all plants in the quadrats with an area of 1 m×1 m2, the relative coverage is the coverage of a plant species divided by the sum of the coverage of all plant species in the quadrats with an area of 1 m×1 m2, and the relative biomass is the biomass of a particular plant species divided by the sum of the biomass of all plant species in the quadrats with an area of 1 m×1 m2.

So, formula (1) is correct.

Lines 204-205: I see formula (5) includes the multiplication sign (×) within each term of this sum. That means this formula should be verbally described as “sum of products of the heights of all plants in the community and their relative biomass, …” (In mathematics, a result of multiplication is called “product”). Whereas your phrase “Plant community height (CWMheight) is the sum of the heights of all plants in the community and their relative biomass” implies that height was summed with mass, and this operation is incorrect because of different dimensionality of the two parameters (relative biomass has no measure unit, whilst height do has it).

Reply:Plant community height (CWMheight) is the sum of the maximum height of each plant in the community multiplied by its relative biomass.

Line 208: Here, it would be useful to refer explanation to formula (1) about calculation of relative biomass. At this I hope, this explanation will be improved to become clearer, then its current version.

Reply:I have made revisions as per the comments of the reviewers

Additional note to style:

第51行:“生物”一词在这里是不必要的。地衣总是生物的。:)也许你的意思是“活地衣”?通常,简单的“表面地衣”将是合适的。

回复:我已根据审稿人的意见进行了修改。
